# Cholesterol promotes *IFNG* mRNA expression in CD4[+] effector/memory cells by SGK1 activation

Aurélie Hanin[1,2,3] ⓘ, Michela Comi[1], Tomokazu S Sumida[1], David A Hafler[1]

**IFNγ-secreting T cells are central for the maintenance of immune surveillance within the central nervous system (CNS). It was previously reported in healthy donors that the T-cell environment in the CNS induces distinct signatures related to cytotoxic capacity, CNS trafficking, tissue adaptation, and lipid homeostasis. These findings suggested that the CNS milieu consisting predominantly of lipids mediated the metabolic conditions leading to IFNγ-secreting brain CD4 T cells. Here, we demonstrate that the supplementation of CD4[+]CD45RO[+]CXCR3[+] cells with cholesterol modulates their function and increases *IFNG* expression. The heightened *IFNG* expression was mediated by the activation of the serum/glucocorticoid-regulated kinase (SGK1). Inhibition of SGK1 by a specific enzymatic inhibitor significantly reduces the expression of *IFNG*. Our results confirm the crucial role of lipids in maintaining T-cell homeostasis and demonstrate a putative role of environmental factors to induce effector responses in CD4[+] effector/memory cells. These findings offer potential avenues for further research targeting lipid pathways to modulate inflammatory conditions.**

## Introduction

T cells are involved in immune surveillance of the central nervous system (CNS) in recognizing microbial infections as blocking T-cell traffic to the CNS that can result in viral, bacterial, and malignant diseases (Ousman & Kubes, 2012). It has more recently been observed that T cells maintain immune surveillance and a distinctive immunological environment within the CNS (Pappalardo et al, 2020). The infiltration of peripheral T cells into the CNS (Smolders et al, 2018) is facilitated by the T helper 1 (Th1)–associated chemokine receptor CXCR3 (Christensen et al, 2004). The distinctive immunological environment within the CNS is necessary, arising from the dual requirement to prevent excessive immune responses while enabling effective surveillance against microbial infections (Ousman & Kubes, 2012).

Single-cell RNA-sequencing analysis of T cells in the cerebrospinal fluid (CSF) of healthy individuals revealed a T-cell IFNγ signature, characterized by the high expression of co-inhibitory receptors that enable cytokine secretion without inducing cell cycle entry (Pappalardo et al, 2020). Moreover, T cells in the CSF exhibit distinct transcriptional profiles compared with those in the blood, reflecting their unique roles in cytotoxicity, CNS trafficking, and tissue adaptation (Pappalardo et al, 2020). Notably, CSF IFNγ T cells demonstrate enrichment in genes associated with cholesterol biosynthesis, suggesting a potential interplay between metabolic factors and IFNγ T-cell homeostasis, possibly involving engagement in a lipid biosynthesis program (Pappalardo et al, 2020).

The brain is the most cholesterol-rich organ, containing ~20% of the body's total cholesterol (Björkhem & Meaney, 2004). Moreover, cholesterol is vital for brain development, synaptogenesis, and neuronal activity, regulating processes such as voltage-dependent and ligand-gated ion channel activity (Björkhem & Meaney, 2004; Dietschy & Turley, 2004; Levitan et al, 2014; Korinek et al, 2015). Cholesterol within the plasma membrane is also crucial for maintaining membrane fluidity and lipid raft formation (Korade & Kenworthy, 2008), influencing the conformation of CD3, an integral component of the TCR complex (Jury et al, 2007; Wu et al, 2015; Guo et al, 2017). Consequently, alterations in lipid content can affect T-cell function by modulating the colocalization of TCR in the membrane (Bietz et al, 2017; Guo et al, 2017; Mailer et al, 2017). Similarly, elevated membrane cholesterol levels in lymphocytes can promote their differentiation into Th1, thereby skewing the immune response toward inflammation (Surls et al, 2012). The sterol response element–binding protein (primarily SREBP-2, with a lesser role of SREBP-1a), responsible for maintaining cholesterol homeostasis, also plays a role in controlling the antigen-driven clonal expansion of CD8[+] T cells (Horton et al, 2002; Kidani et al, 2013).

The discovery of an IFNγ signature in T cells within the CNS prompted us to examine the roles of IFNγ T cells in the CNS. Experiments conducted in animal models demonstrated that T cells, through IFNγ, may contribute to neurodevelopment processes (Yoshida et al, manuscript submitted). Our comprehensive transcriptomic analysis of mouse brains revealed that brain T cells exhibit a robust secretion of IFNγ and express tissue residence

[1]Departments of Neurology and Immunobiology, Yale University School of Medicine, New Haven, CT, USA   [2]Sorbonne Université, Institut du Cerveau—Paris Brain Institute—ICM, Inserm, CNRS, APHP, Hôpital de la Pitié-Salpêtrière, Paris, France   [3]AP-HP, Epilepsy Unit and Clinical Neurophysiology Department, DMU Neurosciences, Hôpital de la Pitié-Salpêtrière, Paris, France

Correspondence: aurelie.hanin@icm-institute.org; david.hafler@yale.edu

proteins necessary for their retention within the brain. Further characterization unveiled a striking transcriptional resemblance between brain CD4 T cells and adipose CD4 T cells, implying brain T cells are initially primed in peripheral tissues such as white adipose and gastrointestinal tissues before trafficking into the brain (Yoshida et al, manuscript submitted). These findings suggest the existence of a novel homeostasis axis between adipose tissue and the brain, where alterations in lipid composition may potentially modulate the activity of brain IFNγ-secreting CD4 T cells to maintain CNS homeostasis.

Previous findings suggest that targeting lipids could offer a new avenue for regulating immune responses. Indeed, studies have demonstrated that culturing T cells with atorvastatin can normalize T-cell signaling (Jury et al, 2006), whereas the use of methyl-β-cyclodextrin, a cholesterol-depleting reagent, can disrupt lipid rafts, prevent TCR clustering, and reverse the hyperactivity of T cells (Krishnan et al, 2004; Deng & Tsokos, 2008). In mice, inhibition of cholesterol synthesis has been shown to regulate Th1/Th2 polarization, favoring the development of Th2 cells (Hakamada-Taguchi et al, 2003). These insights highlight the complex interplay between cholesterol metabolism, lipid rafts, and T-cell function. However, the specific mechanisms through which lipids might affect CNS Th1/IFNγ functions remain elusive and warrant further investigation. Here, we examined human CXCR3$^+$ CD4 memory T cells isolated from the peripheral blood of healthy donors to determine whether and how cholesterol plays a role in driving Th1 homeostasis, including the expression of IFNγ.

# Results

### Impact of the supplementation of cholesterol on the T-cell phenotype

To explore the influence of supplementation of cholesterol on shaping the phenotype of CD4 memory T cells, we cultured CD4$^+$ effector/memory T cells in Xvivo media with or without cholesterol stimulating with anti-CD3/CD28 beads (Fig S1A). Given that CXCR3 is the primary chemokine receptor expressed by CSF and brain T cells, we sorted CXCR3$^+$ CD4 memory T cells from healthy donors (Fig S1A). In contrast to the cells cultured in a high concentration (1:2 bead:cell ratio) of CD3/CD28 beads (positive control), CD4$^+$ effector/memory T cells cultured in cholesterol-supplemented Xvivo media did not enter the cell cycle, even when stimulated with a low concentration (1:50 bead:cell ratio) of CD3/CD28 beads (Fig 1A and B). However, unstimulated cells cultured in cholesterol-supplemented media exhibited a higher expression of CD25 after 72 h compared with cells cultured without cholesterol (%CD4$^+$CD25$^+$ 4.73% ± 2.46% versus 2.62% ± 1.53%, P = 0.003; geometric mean MFI CD4$^+$CD25$^+$ 517.3 ± 139.6 versus 378.4 ± 98.1, P = 0.0024), suggesting a higher activation in the presence of cholesterol (Bajnok et al, 2017) (Fig 1C).

We then used qRT-PCR to assess mRNA expression for cytokine genes. Culturing CXCR3$^+$ memory T cells with cholesterol resulted in a significant increase in the expression of IFNG and TNF, peaking after 12 h of incubation (Fig 1D). Conversely, no increase was detected for the transcription factor TBX21 (T-bet, Fig 1D). The relative

expressions of IFNG and TNF were assessed for the 11 donors at 12 h, revealing an average increase of 94-fold for IFNG and 4.7-fold for TNF in the presence of cholesterol (Fig 1E). We confirmed that cholesterol did not affect the other T helper cell function, given that IL4 and IL17A expressions, along with those of their transcriptional factors GATA3 and RORC, remained unchanged in the presence of cholesterol (Fig S1B). Moreover, neither IL4 nor IL17A mRNAs were detected in unstimulated cells.

Although in the absence of stimulation CD4$^+$ effector/memory T cells did not produce cytokines, even when supplemented with cholesterol, a trend toward a higher proportion of CD4$^+$IFNγ$^+$ cells in the presence of cholesterol (average 2.5-fold increase, Fig 1F) was observed when the cells were stimulated for 72 h with a low concentration (1:50 bead:cell ratio) of CD3/CD28 beads. Notably, the trend toward an increase in the proportion of CD4$^+$IFNγ$^+$ cells was not associated with a higher concentration of IFNγ in the supernatant at 72 h (Fig 1G). However, a significantly higher concentration of TNFα was measured in the supernatant of the cells cultured with cholesterol (median 118.4 ± 63.0 pg/ml versus 85.1 ± 54.1 pg/ml, P = 0.046, Fig 1G).

### Impact of methyl-β-cyclodextrin

To differentiate the effects of cholesterol from those of the methyl-β-cyclodextrin (MBCD), and to assess the role of cholesterol in plasma membranes in regulating Th1 function, we cultured CD4$^+$ effector/memory cells with or without MBCD, a well-established cyclic oligosaccharide commonly used to deplete cholesterol from cellular membranes. The IFNG expression remained unchanged in the presence of MBCD (P = 0.81), whereas a 45-fold increase in IFNG expression was observed in the presence of cholesterol (Fig 2A). Similarly, there was no difference in the concentrations of IFNγ or TNFα in the supernatant of the cells at 72 h (Fig 2B).

### Bulk RNA-sequencing analysis: the underlying mechanisms

As the increase in IFNG mRNA was not linked to an increase in TBX21 mRNA, we performed bulk RNA sequencing to investigate the underlying mechanisms for the modified Th1/IFNγ phenotype in the presence of cholesterol. The 12-h time point was chosen for bulk RNA-sequencing analysis as the IFNG and TNF mRNAs peaked at this time (Fig 1D). A principal component analysis revealed distinct clusters for cells cultured with and without cholesterol (Fig S1C). The differentially expressed genes were identified using the following criteria: a 2-logFC cutoff greater than 2 or less than −2, and a P-value less than 0.05. Differential gene expression analysis identified 327 up-regulated genes including IFNG (fold change 6.23, P < 0.001), JUNB (fold change 2.27, P < 0.001), and SGK1 (fold change 4.34, P < 0.001), whereas 127 genes were down-regulated (Fig 3A). The full list of genes is available in Table S1.

Gene ontology enrichment analysis was performed on differentially expressed genes to add a biological context. Several enriched pathways were identified, including the transcription factor AP-1 complex (enrichment score 13.9, FDR < 0.001), cellular response to cytokine stimulus (enrichment score 11.2, FDR = 0.0015), and positive regulation of leukocyte migration (enrichment score 9.6, FDR = 0.0054) (the full list is available in Table S2). Confirmation

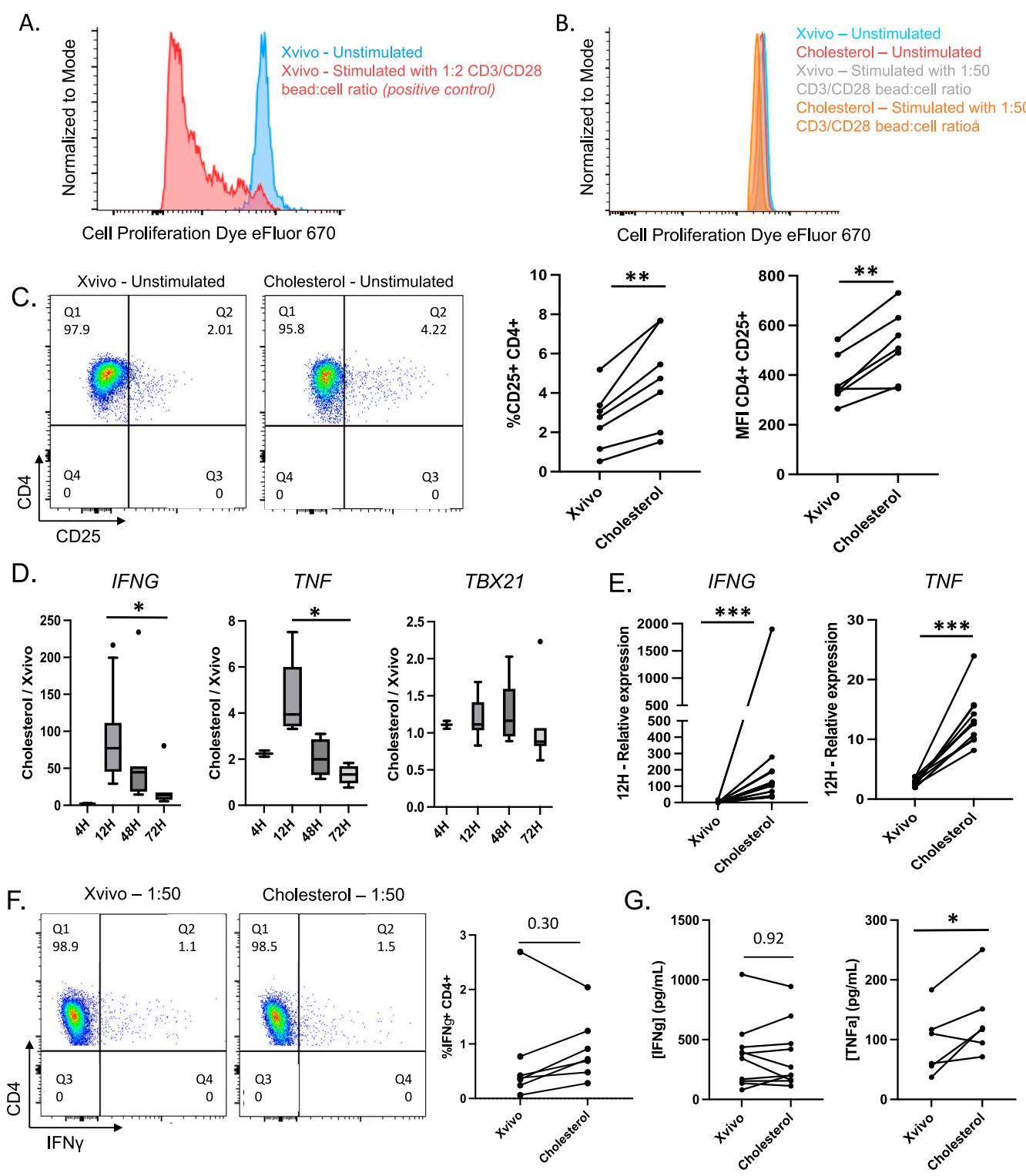

**Figure 1. Impact of cholesterol on the T-cell phenotype.**
**(A)** Representative example of the proliferation of CD4 T cells cultured in Xvivo media for 72 h either unstimulated (blue) or in the presence of a high concentration of CD3/CD28 beads (1:2 bead:cell ratio) (red). **(B)** Representative example of the proliferation of CD4 T cells cultured for 72 h unstimulated in Xvivo media (blue), Xvivo media supplemented by cholesterol (red), Xvivo media with the incorporation of a low concentration of CD3/CD28 beads (1:50 bead:cell ratio) (gray), or Xvivo media supplemented by cholesterol with the incorporation of a low concentration of CD3/CD28 beads (1:50 bead:cell ratio) (orange). **(C)** Representative density plot of CD25 expression in CD4 T cells unstimulated after 72 h of culture with or without cholesterol. The graphs show the increase in percentage and geometric mean (MFI) for CD4+CD25+ cells in the presence of cholesterol (n = 7 donors). **(D, E)** CXCR3+ memory T cells were cultured unstimulated with or without cholesterol from 4 to 72 h. Gene expression was measured by qRT-PCR and normalized to *B2M*. Fig 1D depicts the evolution of the ratio between Xvivo with cholesterol versus Xvivo without cholesterol

of increased gene expressions at 12 h was obtained via qRT-PCR on additional donors (n = 8), showing significant up-regulation of *SGK1* (average 11-fold increase), *FOSB* (average 166.5-fold increase), and *JUNB* (average 2.3-fold increase) (Fig 3B).

### The influence of SGK1 on cholesterol activity

Given the previously described role of SGK1 in Th17 differentiation (Kleinewietfeld et al, 2013) and FOXP3+ Treg function (Hernandez et al, 2015), we investigated whether the increased expression of *SGK1* could trigger Th1/IFNγ activation. To explore SGK1's potential role, we inhibited SGK1 enzymatic activity with GSK650394 in cultures with cholesterol for 12 h. As expected, the SGK1 inhibitor did not alter *SGK1* mRNA expression (Fig 4A). However, blocking SGK1 enzyme activity significantly lowered levels of *IFNG* mRNA compared with cells cultured in cholesterol, although values were not reduced to those of cells cultured without cholesterol (Fig 4A). In addition, SGK1 inhibition did not reduce *FOSB* and *JUNB* mRNA levels (Fig 4B).

## Discussion

Together, these data demonstrate that cholesterol supplementation influences the phenotype of CD4 memory T cells, affecting their activation and cytokine expression. Although we observed increased CD25 expression and elevated *IFNG* and *TNF* mRNA levels, the concentration of IFNγ in the supernatant did not significantly increase in cholesterol-enriched media and those of TNFα were only slightly higher. This discrepancy suggests that although cholesterol may enhance the transcriptional activation of *IFNG* and *TNF*, it does not necessarily translate to increased protein production, possibly because of post-transcriptional or translational regulatory mechanisms. Bulk RNA sequencing revealed differential gene expression in pathways including AP-1 and SGK1. SGK1 emerged as a potential contributor, as its inhibition led to a partial decrease in *IFNG* expression with a similar trend observed for *TNF*, suggesting a role of this kinase in modulating the transcriptional response to cholesterol. These findings indicate that cholesterol may play a role in modulating CD4+ effector/memory cell responses, with SGK1 implicated in Th1/IFNγ regulation.

The CNS environment induces distinct T-cell signatures related to cytotoxic capacity, CNS trafficking, tissue adaptation, and lipid homeostasis. Here, we examined the signals potentially modulating IFNγ-secreting brain CD4+ effector/memory cells and demonstrated that cholesterol influences Th1 function by increasing *IFNG* expression in memory CXCR3+ CD4 T cells. The heightened *IFNG* expression was mediated by the activation of SGK1, a kinase implicated in modulating a number of tissue-driven T-cell functions (Sumida et al, 2024). These data demonstrate the crucial role of

lipids in maintaining CD4+ effector/memory cell homeostasis and a putative role of environmental factors, such as cholesterol, in inducing effector responses in CD4+ effector/memory cells.

Interestingly, cholesterol supplementation also led to an increase in *TNF* mRNA levels, accompanied by a slight increase in TNFα concentrations in the supernatant. In addition, the increase in *TNF* mRNA was attenuated when the cells were cultured with an SGK1 inhibitor. These findings suggest that cholesterol, through activation of SGK1, may also enhance TNFα production by promoting the Th1 function (Kleinewietfeld et al, 2013). The increase in TNFα concentrations could also be mediated through the activation of other downstream signaling pathways, such as NF-κB, which plays a pivotal role in cytokine expression (Liu et al, 2017). The involvement of NF-κB underscores the potential mechanistic link between cholesterol metabolism and immune modulation, indicating that cholesterol may act as an upstream regulator of pro-inflammatory responses by influencing key transcription factors.

Beyond its physical impact on lipid rafts, cholesterol is recognized as a crucial regulator of various pathways, including SGK1. Specifically, the activation of the epithelial sodium channel by cyclosporine A has been found to involve SGK1 activation in a cholesterol-dependent manner (Krueger et al, 2009; Wang et al, 2021). This effect was reversed by lovastatin, an inhibitor of cholesterol synthesis (Wang et al, 2021). SGK1, a serine/threonine protein kinase, plays a vital role in the cellular stress response, modulating processes such as proliferation, survival, and apoptosis (Jang et al, 2022). Elevated levels of this gene may contribute to conditions such as hypertension, hyperglycemia, and diabetic nephropathy (Noor et al, 2021). Surprisingly, SGK1 signaling in mice has been shown to limit the magnitude of the Th1 immune response while promoting Th2 differentiation (Heikamp et al, 2014). In contrast, we and the others demonstrated that SGK1 plays a critical role in balancing Th17 differentiation and Treg function. Notably, we demonstrated that high-salt conditions activate SGK1 during cytokine-induced Th17 polarization (Kleinewietfeld et al, 2013). Moreover, we observed that high-salt conditions increased IFNγ secretion in Tregs through SGK1 activation, consequently impairing Treg function (Hernandez et al, 2015; Sumida et al, 2018). Interestingly, blocking IFNγ synthesis or SGK1 activity resulted in the restoration of the Treg suppressive function, suggesting a strong link between SGK1, IFNγ, and Treg function (Dominguez-Villar et al, 2011; Sumida et al, 2018). SGK1 has also recently been implicated in the loss of Treg in human autoimmune diseases where increases in the PRDM1-short isoform drive SGK1 expression.

It was of interest that cholesterol activated SGK1 even in the absence of high-salt conditions, leading to elevated *IFNG* expression in Th1 CD4 memory T cells. This effect was not explained by a higher proportion of cells entering the cell cycle. However, unlike high-salt conditions in Tregs (Hernandez et al, 2015; Sumida et al, 2018), the increased *IFNG* mRNA did not translate into a significant increase in IFNγ secretion. Similar findings were observed when

over time (4 h: n = 2; 12 h: n = 11; 48 h: n = 6; and 72 h: n = 7). Fig 1E highlights the relative expression for the 11 donors at 12 h (paired evaluation). **(F)** Representative density plot of IFNγ expression in CD4 T cells after 72 h of culture with or without cholesterol and with the incorporation of a low concentration of CD3/CD28 beads (1:50 bead:cell ratio). The graphs show the increase in the percentage of CD4+IFNγ+ cells in the presence of cholesterol for six of the seven donors. **(G)** Concentrations of IFNγ and TNFα were measured in the supernatant of the cells after 72 h of culture with or without cholesterol (IFNγ: n = 10; TNFα: n = 6).

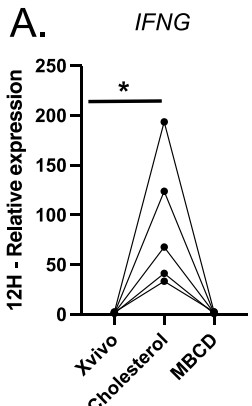

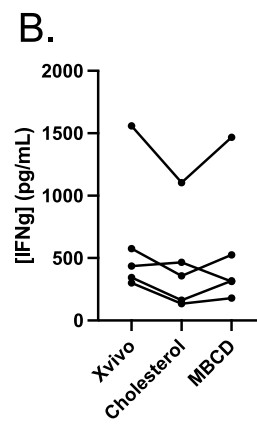

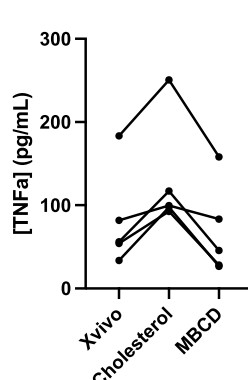

**Figure 2. Impact of methyl-β-cyclodextrin on Th1 functions.**
**(A)** Gene expressions were measured by qRT-PCR and normalized to *B2M* for five donors. Graphs show the up-regulation of *IFNG* with cholesterol, without difference in the presence of MBCD. **(B)** Concentrations of IFNγ and TNFα were measured in the supernatant of the cells after 72 h of culture with or without cholesterol or MBCD (IFNγ: n = 5; TNFα: n = 5).

exposing tissue-resident FOXP3⁺ Tregs to oleic acid and IL-12, suggesting a specific mechanism for lipids (Pompura et al, 2021). The absence of IFNγ secretion despite elevated *IFNG* mRNA levels could be explained by post-transcriptional regulation favored by the enriched cholesterol conditions (Savan, 2014). Indeed, a high-fat diet has been shown to modulate microRNA (miRNA) expression profiles (Karere et al, 2012), with noncoding RNAs, especially miRNAs, emerging as potent regulators of gene expression within T cells (Savan, 2014). Notably, miR-155, positively associated with cholesterol levels (Nemecz et al, 2023) and induced upon T-cell activation, plays a pivotal role in promoting Th1 differentiation by inhibiting IFNγ signaling (Banerjee et al, 2010). Similarly, miR-29 is able to regulate de novo lipogenesis and control innate and adaptive immune responses by targeting *IFNG* (Ma et al, 2011). Conversely, the down-regulation of miR-24 and miR-181 results in an up-regulation of IFNγ secretion by CD4 T cells (Fayyad-Kazan et al, 2014), potentially leading to JAK-STAT activation (Woznicki et al, 2021).

Although the hypothesis that cholesterol modulates Th1 function through SGK1 activation is intriguing, we acknowledge that the data presented are preliminary. Future studies should aim to explore the specific molecular pathways involved, perhaps using more sophisticated techniques such as CRISPR/Cas9-mediated gene editing or specific inhibitors to dissect the roles of cholesterol and SGK1 in greater detail. In addition, further research exploring whether the activation threshold of T cells is dependent on the membrane cholesterol content would be valuable.

Although we observed changes in *IFNG* mRNA levels, further studies are warranted to determine whether these transcriptional changes translate into some functional outcomes, such as cytokine production, T cell–mediated immune responses in vivo, or interaction with other signaling pathways. Functional assays, such as in vitro T-cell activation assays or in vivo models of immune responses, could provide more robust evidence of the biological relevance of these findings. Another potential limitation is the sample size used in this study. Although our data are statistically significant, increasing the population size and examining the biological effects of cholesterol in the context of genetic variation will be of interest.

In summary, culturing CXCR3⁺ memory CD4 T cells in cholesterol-enriched media activates SGK1, leading to increased *IFNG* expression,

which could in turn promote downstream pathways such as JAK-STAT activation (Monteiro et al, 2017; Woznicki et al, 2021). The JAK-STAT pathway could ultimately activate CNS cells, including microglia, resulting in a cytokine storm, cell death, and tissue damage, as demonstrated in COVID-19 (Karki et al, 2021). These findings provide new insights into the potential link between elevated cholesterol concentrations and pro-inflammatory conditions observed in various neurological disorders such as neurodegenerative diseases or status epilepticus (Djelti et al, 2015; Boussicault et al, 2018; Hanin et al, 2021). In addition, the high cholesterol-rich environment in the CNS provides a theoretical framework for the acquisition of the lipid/IFNγ signature associated with T-cell traffic into the CNS.

# Materials and Methods

### Study design

This study sought to assess the influence of cholesterol on the CD4 memory T-cell phenotype. A comprehensive approach was employed, combining in vitro experimental assays with ex vivo computational analyses using human CD4 memory T cells isolated from peripheral blood obtained from healthy donors.

### Study participants

PBMCs were isolated from 11 healthy donors, with an average age of 30 yr (minimum age, 21 yr; maximum age, 58 yr). The specific number of donors used for each analysis is detailed in the legend corresponding to each figure.

### Human T-cell isolation and culturing

PBMCs were isolated from healthy donors using a Lymphoprep gradient centrifugation method (07851; STEMCELL Technologies). Memory CD4⁺ T cells were negatively selected using Memory CD4⁺ T Cell Isolation Kit (19157; STEMCELL Technologies). Subsequently, cells were stained with Cell Proliferation Dye eFluor 670 (65084090; Thermo Fisher Scientific) as per the manufacturer's protocol. After

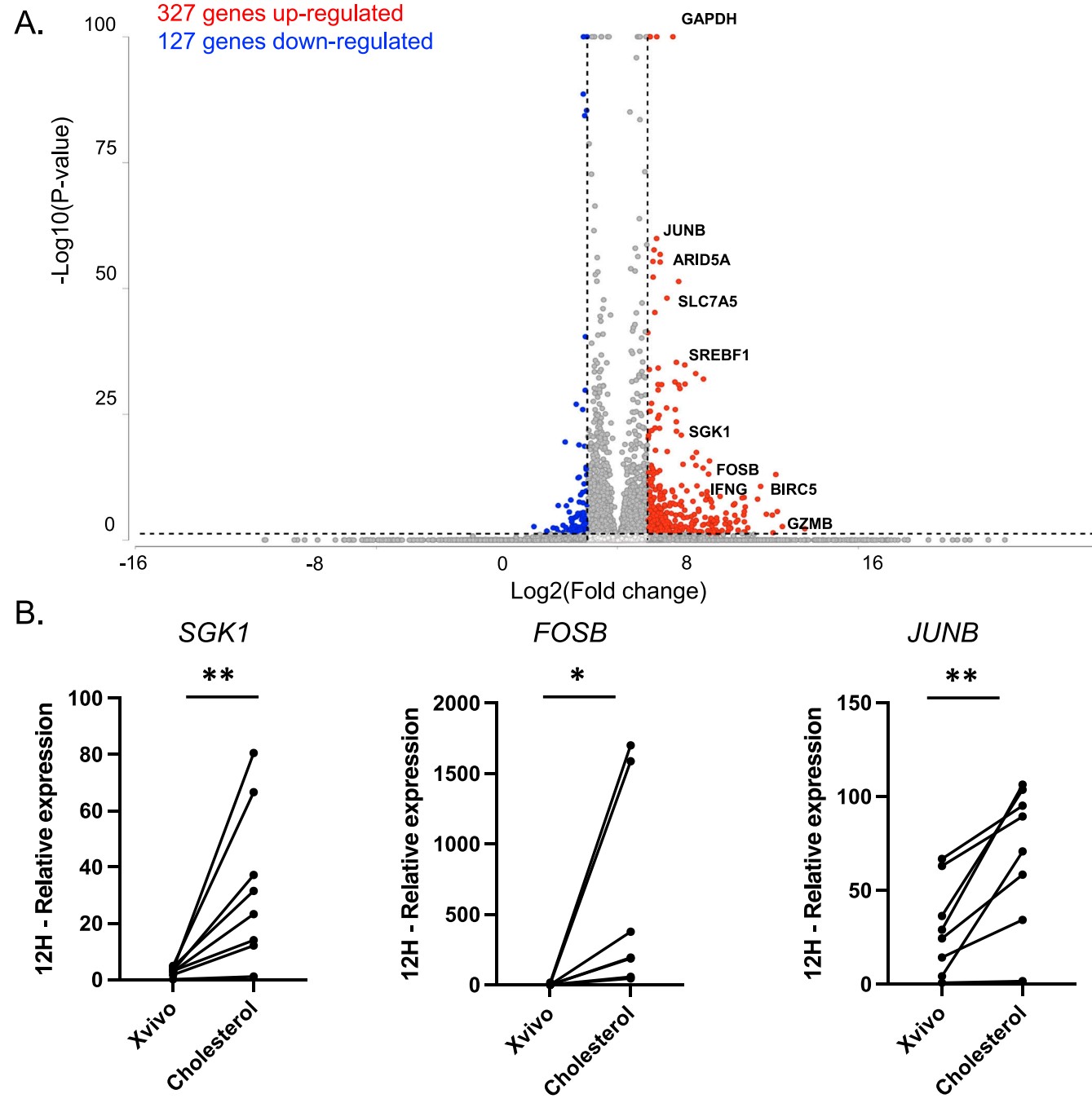

**Figure 3. Bulk RNA-sequencing analysis to unravel the underlying mechanisms for the effect of cholesterol on the T-cell phenotype.**
**(A)** Volcano plots of the genes up-regulated or down-regulated in cells cultured for 12 h in Xvivo media supplemented or not by cholesterol without CD3/CD28 stimulation (n = 3 donors, criteria: absolute 2-fold change greater than 2 and *P*-value less than 0.05). **(B)** Gene expressions were measured by qRT-PCR for *SGK1*, *FOSB*, and *JUNB* and normalized to *B2M* for eight additional donors. Graphs highlight the relative expression at 12 h (paired evaluation).

two washes in PBS to eliminate serum, cells were incubated for 10 min at 37°C in the dark. Staining was halted by adding four volumes of cold fetal bovine serum, followed by three washes with complete media.

For flow cytometry, cells were stained for 30 min at 4°C with antibodies against CD4 (clone RPA-T4, 560650; BD Biosciences), CD8 (clone human CD8a, 562311; BD Biosciences), CD45RO (clone UCHL1,

555492; BD Biosciences), and CD183/CXCR3 (clone 1C6/CXCR3, 562558; BD Biosciences). Fc Block Reagent (564219; BD Biosciences) was incorporated into the staining media to minimize nonspecific antibody staining. CD4$^+$CD8$^-$CD45RO$^{high}$CXCR3$^+$ T cells were sorted using a FACSAria (BD Biosciences) (Fig S1A).

Sorted T cells were cultured at 37°C for 4–72 h in 100 μl of Xvivo medium supplemented with 100 U/μg/ml penicillin/streptomycin

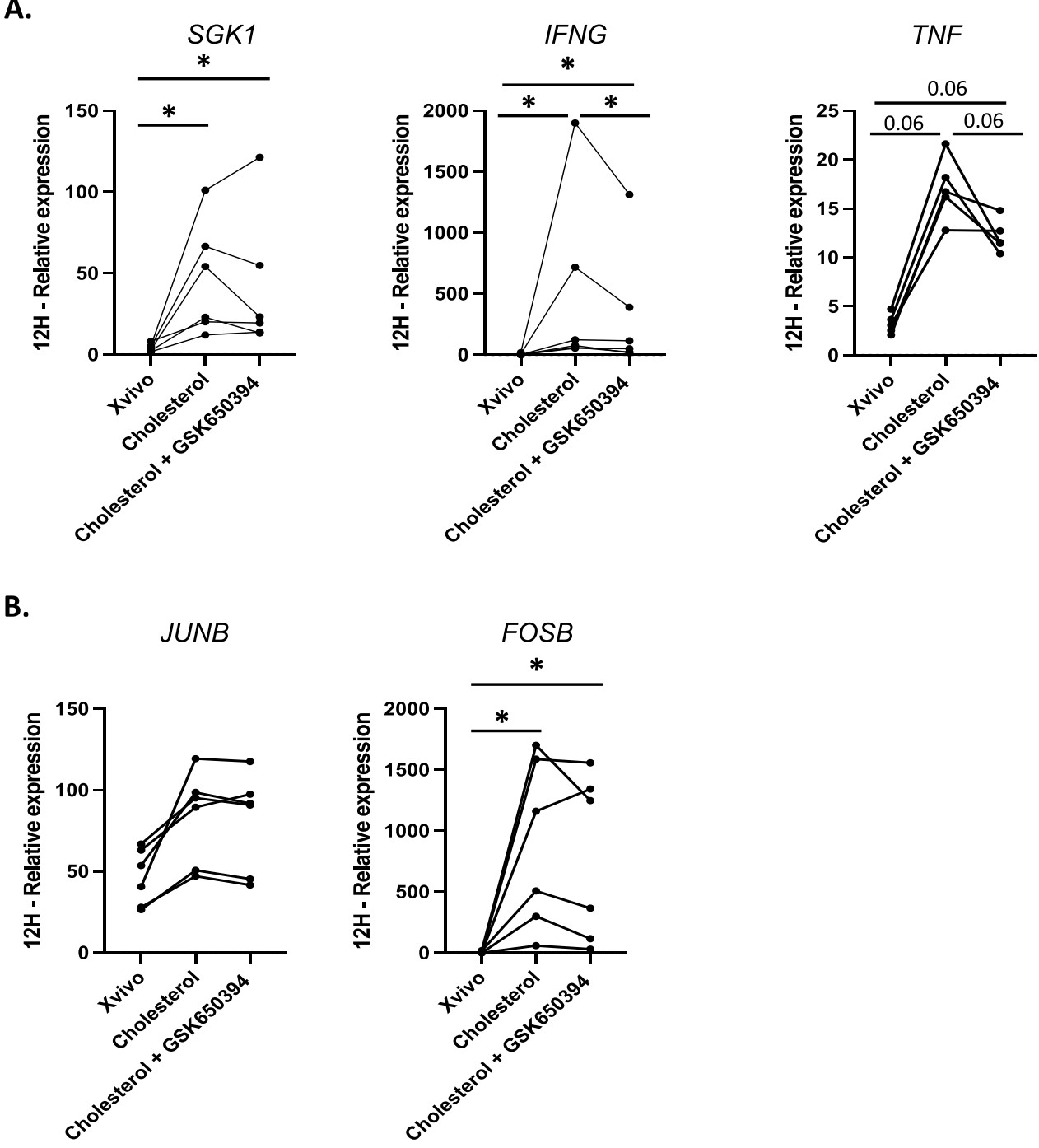

**Figure 4. Influence of SGK1 on cholesterol activity.**
Gene expressions were measured by qRT-PCR and normalized to B2M for six donors (five donors for *TNF*). **(A, B)** Graphs show the down-regulation of *IFNG* and *TNF* mRNA in the presence of SGK1 inhibitor (A), whereas no changes are observed for *JUNB* and *FOSB* (B). Statistical analysis: We initially applied a Kruskal–Wallis test to assess any differences among the three culture conditions (i.e., Xvivo, cholesterol-enriched media, and cholesterol-enriched media + SGK1 inhibitor). Post hoc comparisons were performed using Wilcoxon's tests when appropriate.

(15140; Gibco), with or without 10 µg/ml cholesterol (cholesterol–water soluble, a cholesterol–methyl-β-cyclodextrin complex, diluted in Xvivo medium, C4951; Sigma-Aldrich). To specifically differentiate the effect of cholesterol from that of methyl-β-cyclodextrin, additional experiments were performed where cells were cultured in Xvivo medium either with or without 10 µg/ml cholesterol or with or without 10 µg/ml methyl-β-cyclodextrin (C4555; Sigma-Aldrich). The cells were plated at 50,000 per well in

ninety-six-well round-bottomed plates (3799; Corning Costar). To assess the cells' capacity to synthesize IFNγ, a subset of the cells was stimulated by adding 1,000 Human T-activator CD3/CD28 beads per well (11131D; Thermo Fisher Scientific).

## SGK1 modulation

In select experiments, cells were cultured in the presence or absence of 1 $\mu$M of the specific SGK1 enzymatic inhibitor GSK650394 (Tocris Bioscience, R&D Systems) added to the culture media for 12 h.

## Flow cytometry

Cells were subjected to flow cytometry after a culture period of 72 h. For surface staining, cells were stained with the following antibodies for 15 min at RT: CD4 (clone RPA-T4, 560650; BD Biosciences), CD183/CXCR3 (clone 1C6/CXCR3, 562558; BD Biosciences), PD1 (clone EH12.1, 561272; BD Biosciences), and CD25 (clone M-A251, 560989; BD Biosciences). The LIVE/DEAD fixable 633/780 (L34993; Life Technologies Corporation) was incorporated into the incubation mix to exclude dead cells.

For intracellular staining, cells were exposed to GolgiPlug (B5936; BD Biosciences) for 6 h, fixed with Cytofix (554655; BD Biosciences), permeabilized (Cytofix/CytoPerm, 51-2090KZ; BD Biosciences) following the manufacturer's instructions, and stained with IFNγ antibody (clone 4SB3, 502544; BD Biosciences) for 30 min in permeabilization buffer (Perm/Wash Buffer, 554723; BD Biosciences). To prevent inducing bias, PMA/ionomycin was not added to our culture media.

Data acquisition was performed on LSR II (BD Biosciences) and analyzed using FlowJo software (TreeStar).

## Cytokine measurements

Culture supernatants from stimulated cells were collected at 72 h. IFNγ concentrations were measured using the ELISA IFNγ kit (OptEIA IFNγ, 555142; BD Biosciences), following the manufacturer's instructions. Supernatant concentrations of TNFα were measured using multiplex fluorescent bead–based immunoassay detection (BD Biosciences) with BD Cytometric Bead Array Human Soluble Protein Master Buffer Kit. All samples were measured undiluted, and cytokine concentrations were interpolated from the corresponding calibration curve and expressed in pg/ml.

## qRT-PCR

Cells for RNA isolation were harvested between 4 and 72 h of culture, and total RNA was extracted using RNeasy Micro Kit (QIAGEN). The RNA was then retrotranscripted to cDNA using TaqMan Reverse Transcription Reagents (Applied Biosystems). cDNAs were amplified with TaqMan probes (TaqMan Gene Expression Arrays) and TaqMan Fast Advanced Master Mix on StepOne Real-Time PCR System (Applied Biosystems) following the manufacturer's instructions. mRNA expression was measured relative to $\beta$2-microglobulin expression. Values are normalized to the expression of $\beta$2-microglobulin for each sample using the following

formula: relative RNA expression = $(2^{-\Delta CT})$ x 1,000. For certain analyses, the relative expressions were compared between two conditions and expressed as a ratio using the formula: ratio = $2^{-(\Delta CTcondition1-\Delta CTcondition2)}$.

The following primers and probes were used for TaqMan assays (all from Thermo Fisher Scientific): *B2M* (Hs00187842_m1), *FOSB* (Hs00171851_m1), *GATA3* (Hs00231122_m1), *IFNG* (Hs00989291_m1), *IL17A* (Hs00174383_m1), *IL4* (Hs00174122_m1), *JUNB* (Hs00357891_s1), *RORC* (Hs01076112_m1), *SGK1* (Hs00178612_m1), *TBX21* (Hs00203436_m1), and *TNF* (Hs99999043_m1).

## Bulk RNA-seq library preparation and data analysis

For bulk RNA-seq analysis, CD4$^+$CD8$^-$CD45RO$^{high}$CXCR3$^+$ T cells from three healthy donors were sorted by FACS and cultured without stimulation for 12 h, in the presence or absence of cholesterol at 10 $\mu$g/ml. After harvesting, RNA was isolated using RNeasy Micro Kit (QIAGEN). cDNA was synthesized for bulk RNA-seq using SMART-Seq V4 Ultra Low Input RNA Kit for Sequencing (Clontech, Takara Bio). Barcode libraries were then generated using Nextera XT DNA Library Preparation Kit (Illumina) and subjected to sequencing with a 2 × 100 bp paired-end protocol on NovaSeq (Illumina). The resulting databases were uploaded to Partek software and analyzed following the manufacturer's instructions. The differentially expressed genes were identified using the following criteria: a 2-logFC cutoff greater than 2 or less than −2, and a *P*-value less than 0.05. We next performed gene ontology enrichment analysis (Partek) on the list of genes filtered by the above parameters.

## Statistics

All analyses were conducted with R Studio (v.1.2.5019), and graphical representations were created with GraphPad Prism (GraphPad Software Inc., La Jolla, CA). Significance levels were determined by a paired Student or Wilcoxon test, as appropriate. When three conditions were compared, we initially applied a Kruskal–Wallis test and performed post hoc comparisons using the Wilcoxon test. A significance threshold of $P < 0.05$ was applied with the notation as follows: *$P < 0.05$, **$P < 0.01$, and ***$P < 0.001$.

## Study approval

Approval for all experiments conducted in this study was obtained from the Yale Institutional Review Board (#200002791; Neurology Data Registry & Biorepository). In addition, written informed consent was obtained from all participating donors before their involvement in the study. The research adhered to the ethical principles outlined in the World Medical Association (WMA) Declaration of Helsinki and the US Department of Health and Human Services Belmont Report.

# Data Availability

The data are available from the corresponding author upon request.

# Supplementary Information

# Acknowledgements

The authors thank Yale Flow Cytometry for their assistance with Fortessa345 and FACSAria service. The Core is supported in part by National Cancer Institute (NCI) Cancer Center Support Grant # NIH P30 CA016359. The authors also thank the Yale Center for Genome Analysis for their help with the bulk RNA-sequencing analysis. Research reported in this publication was supported by the National Institute of General Medical Sciences of the National Institutes of Health under Award Number 1S10OD030363-01A1. A Hanin received postdoctoral grants from the Paratonnerre Association, the Servier Institute, the Philippe Foundation, the Swebilius Foundation, and the NORSE/FIRES Research Fund at Yale for status epilepticus–related research.

## Author Contributions

A Hanin: conceptualization, data curation, formal analysis, funding acquisition, validation, investigation, visualization, methodology, and writing—original draft, review, and editing.
M Comi: conceptualization, data curation, supervision, visualization, methodology, and writing—original draft.
TS Sumida: formal analysis, supervision, visualization, and writing—original draft, review, and editing.
DA Hafler: conceptualization, formal analysis, supervision, funding acquisition, validation, visualization, methodology, project administration, and writing—original draft, review, and editing.

## Conflict of Interest Statement

DA Hafler has received research funding from Bristol Myers Squibb, Novartis, Sanofi, and Genentech. He has been a consultant for Bayer Pharmaceuticals, Repertoire Inc, Bristol Myers Squibb, Compass Therapeutics, EMD Serono, Genentech, Juno therapeutics, Novartis Pharmaceuticals, Proclara Biosciences, Sage Therapeutics, and Sanofi. The remaining authors have declared no conflicts of interest.

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
