## [Reviewer comments · Life Science Alliance]

Life Science Alliance

Cholesterol Promotes IFNG mRNA Expression in CD4+ effector/memory Cells by SGK1 Activation

Aurelie Hanin, Michela Comi, Tomokazu Sumida, and David Hafler

DOI: <https://doi.org/10.26508/lsa.202402890>

Corresponding author(s): Aurelie Hanin, Yale University

Review Timeline:

Submission Date:	2024-06-18
Editorial Decision:	2024-07-29
Revision Received:	2024-09-23
Editorial Decision:	2024-09-24
Revision Received:	2024-09-26
Accepted:	2024-09-26

Transaction Report:

July 29, 2024

Re: Life Science Alliance manuscript #LSA-2024-02890-T

Dr. Aurelie Hanin
Yale School of Medicine
300 George St
New Haven, CT 06511

Dear Dr. Hanin,

Thank you for submitting your manuscript entitled "Cholesterol Induces T cell Expression of IFNG mRNA Mediated by SGK1" to Life Science Alliance. The manuscript was assessed by expert reviewers, whose comments are appended to this letter. We invite you to submit a revised manuscript addressing the Reviewer comments.

Thank you for this interesting contribution to Life Science Alliance. We are looking forward to receiving your revised manuscript.

Sincerely,

B. MANUSCRIPT ORGANIZATION AND FORMATTING:

Reviewer #1 (Comments to the Authors (Required)):

Hanin et al describe in this piece of work the effect of cholesterol supplementation in the expression of IFNG in CD4+CD45RO+CXCR3+ cells. Their results show that although frequency and secretion of IFN are not changed, cholesterol supplementation in the media induces an increase in IFNG levels. They further demonstrate the role of SGK1 in partially regulating this process.

The work is original in that it increases our knowledge about the impact that cholesterol has in the regulation of the human effector immune response. It also provides a novel mechanistic association between SGK1 and IFNG through cholesterol supplementation.

I have some comments that, in my opinion, might improve the manuscript.

Major comments:

1. Cholesterol is not quantified in any of the experiments, hence, for accuracy, authors should either note that they supplement their cultures with cholesterol, or actually quantify increased cholesterol content in their cells.
2. Did authors run cholesterol solvent control to rule out any possible effect of cyclodextrin in their system? This should be included in the methods (if done) or at least discussed, if not done.
3. When allocating significance to the RNAseq dataset, q-values, rather than p-values, should be used. Can authors confirm that their significance has been corrected for multiple testing?

Minor comments:

1. Title should be more accurate. It should at least describe that the work refers to human CD4+CD45RO+CXCR3+ cells, and not T cells in general. Also, it is cholesterol supplementation, as there is no evidence in the work that there is an increase in cholesterol content in their cells.

2. Introduction.

- In paragraph 3, authors name SREBP as responsible for the maintenance of cellular cholesterol homeostasis. This should be more accurate and relate to SREBP-2 specifically (i.e. Robinson et al, *Frontiers Immunology* 2017).

- In paragraph 5 authors state that "the impact of cholesterol in the activation of T cells or its impact is predominantly on IFN γ -secreting T cells has not been investigated". However, there are several reports that have investigated the role of cholesterol itself and cholesterol metabolism in the Th1 response, in both human and mouse, making this sentence not fully accurate. Some examples of these are: Surls et al, *Plos One* 2012; Hakamada-Taguchi et al, *Circulation Research* 2003; Perucha et al, *Nat Commun* 2019.

2. Results

For accuracy, instead of T cells, authors should refer to CD4+ effector/memory or CD45RO+CXCR3+ cells.

3. Discussion

- The work of Heikamp et al (*Nature Immunology* 2014), where they describe the role of SGK1 in promoting Th2 responses while blocking Th1 is not mentioned, but it seems really relevant in the context of this work and should be included.

- Discussion seems somewhat overstated. Authors do not show that cholesterol modulates Th1 function (IFN levels are not changed in their system), but they show that IFN mRNA levels are increased.

Reviewer #2 (Comments to the Authors (Required)):

Hanin et al have investigated the effect of cholesterol on T cell function. The authors have sorted CXCR3+CD4+ T cells from healthy donors and treated them with cholesterol in vitro.

They find that cholesterol exposure leads to a subtle state of T cell activation, with upregulation of IFNG and TNF mRNA.

Cholesterol-treated T cells did not enter the cell cycle, but there was a small increase in the frequency of CD25+ T cells. Cholesterol treated cells did not secrete IFN γ but did release TNF. As expected, the cholesterol effect was blocked with a pharmacological inhibitor of SGK1.

Comments:

1. While the hypothesis is interesting, the data presented are very preliminary and only partially convincing. The main issue with this study is the lack of any mechanistic data and the lack of demonstrating biological significance of the observation.
2. Sample numbers are too small.
3. In some data (eg, Figure 1F), the effects appear quite subtle, and it is unclear whether such small changes could have any functional significance.
4. While the authors observed increased CD25 expression and elevated IFN γ mRNA, the concentration of IFN γ in the supernatant did not significantly increase in cholesterol-enriched media (Discussion line 182-184). This questions whether there is any functional impact of the cholesterol effect.
5. Not only did cholesterol in the media induce IFN γ mRNA, it also induced TNF mRNA and there was some detectable TNF in the supernatant. TNF is insufficiently considered in this manuscript.
6. The PCA plot in Supplement Figure 1C is not convincing. In both groups, two of the three samples are very similar, but one sample differs significantly from the other two in gene expression, making the validity of the analysis questionable.
7. That SGK1 mediates the effects of membrane cholesterol has long been known (PMID: 19910701). Do the authors know to which extent the cholesterol in the medium alters membrane composition and to which extent it is metabolized by the cells?
8. Cholesterol treatment induces a subtle state of activation (CD25, IFN γ mRNA, TNF mRNA). Is the activation threshold of a T cells dependent on the membrane cholesterol content? The authors should provide data supporting this.

Reviewer #1 (Comments to the Authors (Required)):

Hanin et al describe in this piece of work the effect of cholesterol supplementation in the expression of IFNG in CD4⁺CD45RO⁺CXCR3⁺ cells. Their results show that although frequency and secretion of IFNG are not changed, cholesterol supplementation in the media induces an increase in IFNG levels. They further demonstrate the role of SGK1 in partially regulating this process.

The work is original in that it increases our knowledge about the impact that cholesterol has in the regulation of the human effector immune response. It also provides a novel mechanistic association between SGK1 and IFNG through cholesterol supplementation.

I have some comments that, in my opinion, might improve the manuscript.

Major comments:

1. Cholesterol is not quantified in any of the experiments, hence, for accuracy, authors should either note that they supplement their cultures with cholesterol, or actually quantify increased cholesterol content in their cells.

REPLY: We agree with the reviewer. We edited the manuscript as follows:

Abstract: "that the supplementation of CD4⁺CD45RO⁺CXCR3⁺ cells with cholesterol modulates their function and increases *IFNG* expression". We now use the word supplement throughout the paper.

2. Did authors run cholesterol solvent control to rule out any possible effect of cyclodextrin in their system? This should be included in the methods (if done) or at least discussed, if not done.

REPLY: We appreciate the reviewer's insightful question. It is indeed important to note that the cholesterol-water soluble used in our experiments is a complex of cholesterol and methyl- β -cyclodextrin, a well-established cyclic oligosaccharide that enhances cholesterol solubility. To address potential confounding effects, we also examined the individual impact of the methyl- β -cyclodextrin alone, as it is commonly employed to deplete cholesterol from the cellular membrane when used independently. The method was updated as follows:

"Sorted T cells were cultured at 37°C for 4 to 72 hours in 100 μ L of Xvivo medium supplemented with 100 U/ μ g/mL penicillin/streptomycin (Gibco, 15140), with or without 10 μ g/mL cholesterol (cholesterol-water soluble, a cholesterol-methyl- β -cyclodextrin complex, diluted in Xvivo medium, Sigma-Aldrich C4951). To specifically differentiate the effect of cholesterol from those of methyl- β -cyclodextrin, additional experiments were performed where cells were cultured in Xvivo medium either with or without 10 μ g/mL cholesterol or with or without 10 μ g/mL methyl- β -cyclodextrin (Sigma-Aldrich C4555)."

Results were added on page 5 and Figure 2, as follows:

Impact of methyl- β -cyclodextrin

To differentiate the effects of cholesterol from those of the methyl- β -cyclodextrin (mbcd), and to assess the role of cholesterol in plasma membranes in regulating Th1 function, we cultured CD4⁺ effector/memory cells with or without mbcd, a well-established cyclic oligosaccharide commonly used to deplete cholesterol from cellular membranes. The *IFNG* expression

remained unchanged in the presence of mbcD ($p=0.81$), whereas a 45-fold increase in *IFNG* expression was observed in the presence of cholesterol (Figure 2A). Similarly, there was no difference in the concentrations of $IFN\gamma$ or $TNF\alpha$ in the supernatant of the cells at 72 hours (Figure 2B).

3. When allocating significance to the RNAseq dataset, q-values, rather than p-values, should be used. Can authors confirm that their significance has been corrected for multiple testing?

REPLY: We used FDR instead of q-values and had provided the FDRs in Supplementary Table 2, specifically in column F. The results and discussion sections have been revised accordingly:

- On page 6: Several enriched pathways were identified, including the transcription factor AP-1 complex (enrichment score 13.9, $FDR<0.001$), cellular response to cytokine stimulus (enrichment score 11.2, $FDR=0.0015$), and positive regulation of leukocyte migration (enrichment score 9.6, $FDR=0.0054$) (the full list is available in Supplementary Table 2).
- The JAK-STAT pathway has been removed from the results and discussion as the p-value was no longer significant after correction for multiple tests.

Minor comments:

1. Title should be more accurate. It should at least describe that the work refers to human CD4+CD45RO+CXCR3+ cells, and not T cells in general. Also, it is cholesterol supplementation, as there is no evidence in the work that there is an increase in cholesterol content in their cells.

REPLY: The title was edited as: "Cholesterol promotes *IFNG* mRNA Expression in CD4⁺ effector/memory cells by SGK1 activation".

We prefer not to use the word "supplementation" in the title because of its dual meaning as a dietary "supplement" and explain this in detail in the abstract.

2. Introduction.

- In paragraph 3, authors name SREBP as responsible for the maintenance of cellular cholesterol homeostasis. This should be more accurate and relate to SREBP-2 specifically (i.e. Robinson et al., Frontiers Immunology 2017).

REPLY: Although SREBP-2 is the primary SREBP responsible for maintaining cellular cholesterol homeostasis, SREBP-1a is a powerful activator of all SREBP-responsive genes, including those involved in cholesterol synthesis.

We edited the introduction as follows:

"The sterol response element-binding protein (primarily SREBP-2, with a lesser role for SREBP-1a), responsible for maintaining cholesterol homeostasis, also plays a role in controlling antigen-driven clonal expansion of CD8⁺ T cells (Kidani et al., 2013; Horton et al., 2002)."

We added the reference:

"Horton, J.D., J.L. Goldstein, and M.S. Brown. 2002. SREBPs: activators of the complete program of cholesterol and fatty acid synthesis in the liver. *J Clin Invest.* 109:1125–1131. doi:10.1172/JCI15593."

- In paragraph 5 authors state that "the impact of cholesterol in the activation of T cells or

its impact is predominantly on IFN γ -secreting T cells has not been investigated". However, there are several reports that have investigated the role of cholesterol itself and cholesterol metabolism in the Th1 response, in both human and mouse, making this sentence not fully accurate. Some examples of these are: Surls et al, Plos One 2012; Hakamada-Taguchi et al, Circulation Research 2003; Perucha et al, Nat Commun 2019.

REPLY: We apologize for the oversight. The introduction has been revised as follows:

- Paragraph 3: "Consequently, alterations in lipid content can affect T cell function by modulating the colocalization of TCR in the membrane (Guo et al., 2017; Bietz et al., 2017; Mailer et al., 2017). Similarly, elevated membrane cholesterol levels in lymphocytes can promote their differentiation into Th1, thereby skewing the immune response towards inflammation (Surls et al., 2012)."
- Paragraph 5: "In mice, inhibition of cholesterol synthesis has been shown to regulate Th1/Th2 polarization, favoring the development of Th2 cells (Hakamada-Taguchi et al., 2003). These insights highlight the complex interplay between cholesterol metabolism, lipid rafts, and T-cell function. However, the specific mechanisms through which lipids might affect CNS Th1/IFN γ functions remain elusive and warrant further investigation. Here, we examined human CXCR3⁺ CD4 memory T cells isolated from the peripheral blood of healthy donors to determine whether and how cholesterol plays a role in driving Th1 homeostasis, including the expression of IFN γ ."

2. Results

For accuracy, instead of T cells, authors should refer to CD4⁺ effector/memory or CD45RO⁺CXCR3⁺ cells.

REPLY: We edited the manuscript as suggested and included this in the title.

3. Discussion

- The work of Heikamp et al (Nature Immunology 2014), where they describe the role of SGK1 in promoting Th2 responses while blocking Th1 is not mentioned, but it seems really relevant in the context of this work and should be included.

REPLY: We now discussed this paper in the discussion as follows:

"Elevated levels of this gene may contribute to conditions such as hypertension, hyperglycemia, and diabetic nephropathy (Noor et al., 2021). Surprisingly, SGK1 signaling in mice has been shown to limit the magnitude of the Th1 immune response while promoting Th2 differentiation (Heikamp et al., 2014). In contrast, we and the others demonstrated that SGK1 plays a critical role balancing Th17 differentiation and Treg function. Notably, we demonstrated that high-salt conditions activate SGK1 during cytokine-induced Th17 polarization (Kleinewietfeld et al., 2013)."

- Discussion seems somewhat overstated. Authors do not show that cholesterol modulates Th1 function (IFN γ levels are not changed in their system), but they show that IFN γ mRNA levels are increased.

REPLY: We revised the discussion as follows:

- Together, these data demonstrate that cholesterol supplementation influences the phenotype of CD4 memory T cells, affecting their activation and cytokine expression. While we observed increased CD25 expression and elevated *IFNG* and *TNF* mRNA levels, the concentration of IFN- γ in the supernatant did not significantly increase in

cholesterol-enriched media and those of $TNF\alpha$ were only slightly higher. This discrepancy suggests that while cholesterol may enhance the transcriptional activation of *IFNG* and *TNF*, it does not necessarily translate to increased protein production, possibly due to post-transcriptional or translational regulatory mechanisms. Bulk RNA sequencing revealed differential gene expression in pathways including AP-1 and SGK1. SGK1 emerged as a potential contributor, as its inhibition led to a partial decrease in *IFNG* expression with a similar trend observed for *TNF*, suggesting a role for this kinase in modulating the transcriptional response to cholesterol. These findings indicate that cholesterol may play a role in modulating $CD4^+$ effector/memory-cell responses, with SGK1 implicated in Th1/IFN-g regulation.

- The CNS environment induces distinct T-cell signatures related to cytotoxic capacity, CNS trafficking, tissue adaptation, and lipid homeostasis. Here, we examined the signals potentially modulating $IFN\gamma$ -secreting brain $CD4^+$ effector/memory cells and demonstrated that cholesterol influences Th1 function by increasing *IFNG* expression in memory $CXCR3^+$ $CD4$ T cells. The heightened *IFNG* expression was mediated by the activation of SGK1, a kinase implicated in modulating a number of tissue driven T cell functions (Sumida et al., 2024). These data demonstrate the crucial role of lipids in maintaining $CD4^+$ effector/memory cell homeostasis and a putative role for environmental factors, such as cholesterol, in inducing effector responses in $CD4^+$ effector/memory cells.
- In summary, culturing $CXCR3^+$ memory $CD4$ T cells in cholesterol-enriched media activates SGK1, leading to increased *IFNG* expression, which could in turn promote downstream pathways such as JAK-STAT activation (Monteiro et al., 2017; Woznicki et al., 2021).

Reviewer #2 (Comments to the Authors (Required)):

Hanin et al have investigated the effect of cholesterol on T cell function. The authors have sorted CXCR3+CD4+ T cells from healthy donors and treated them with cholesterol in vitro.

They find that cholesterol exposure leads to a subtle state of T cell activation, with upregulation of IFNG and TNF mRNA.

Cholesterol-treated T cells did not enter the cell cycle, but there was a small increase in the frequency of CD25+ T cells.

Cholesterol treated cells did not secrete IFN γ but did release TNF. As expected, the cholesterol effect was blocked with a pharmacological inhibitor of SGK1.

Comments:

1. While the hypothesis is interesting, the data presented are very preliminary and only partially convincing. The main issue with this study is the lack of any mechanistic data and the lack of demonstrating biological significance of the observation.

2. Sample numbers are too small.

REPLY to 1 and 2: We acknowledge the limitations of our study, including the preliminary nature of the data and the need for further mechanistic insights. In this paper, we focused mostly on biological investigation. However, we did find that the downregulation of SGK1 reverses the effect of SGK1 providing valuable initial evidence of the potential role of cholesterol in modulating Th1 function through SGK1.

The limitations have been added to the discussion as follows:

“While the hypothesis that cholesterol modulates Th1 function through SGK1 activation is intriguing, we acknowledge that the data presented are preliminary. Future studies should aim to explore the specific molecular pathways involved, perhaps using more sophisticated techniques such as CRISPR/Cas9-mediated gene editing or specific inhibitors to dissect the roles of cholesterol and SGK1 in greater detail. Additionally, further research exploring whether the activation threshold of T cells is dependent on the membrane cholesterol content would be valuable.

While we observed changes in *IFNG* mRNA levels, further studies are warranted to determine whether these transcriptional changes translate into some functional outcomes, such as cytokine production, T cell-mediated immune responses in vivo, or interaction with other signaling pathways. Functional assays, such as in vitro T cell activation assays or in vivo models of immune responses, could provide more robust evidence of the biological relevance of these findings. Another potential limitation is the sample size used in this study. While our data are statistically significant, increasing the population size and examining the biological effects of cholesterol in the context of genetic variation will be of interest.”

Despite these limitations, we consider that the insights gained from this study are important to report, as they lay the groundwork for future research that could further elucidate these mechanisms and their implications for immune regulation.

3. In some data (eg, Figure 1F), the effects appear quite subtle, and it is unclear whether such small changes could have any functional significance.

REPLY: We appreciate the reviewer's observation regarding the subtle effects observed in some data, such as in Figure 1F regarding the relative higher proportion of CD4⁺IFN γ ⁺ cells in the

presence of cholesterol. However, we want to highlight that the increase in *IFNG* mRNA and *SGK1* mRNA are important and consistent throughout experiments.

While some other changes may appear modest, it is important to recognize that even small alterations in gene expression or protein levels can have significant downstream effects, particularly in the context of complex biological systems like immune cell signaling. These subtle changes may act in a cumulative or synergistic manner, leading to more pronounced functional outcomes that are not immediately apparent in isolated assays. However, we acknowledge that further investigation is needed to determine whether these subtle changes translate into meaningful functional effects *in vivo* or under physiological conditions. Future studies, particularly those involving more sensitive assays or *in vivo* models, will be essential to explore the functional significance of these observations.

We added this limitation in the discussion as follows:

“While we observed changes in *IFNG* mRNA levels, further studies are warranted to determine whether these transcriptional changes translate into some functional outcomes, such as cytokine production, T cell-mediated immune responses *in vivo*, or interaction with other signaling pathways. Functional assays, such as *in vitro* T cell activation assays or *in vivo* models of immune responses, could provide more robust evidence of the biological relevance of these findings.”

4. While the authors observed increased CD25 expression and elevated IFNG mRNA, the concentration of INF γ in the supernatant did not significantly increase in cholesterol-enriched media (Discussion line 182-184). This questions whether there is any functional impact of the cholesterol effect.

REPLY: We agree with the reviewer that the lack of a significant increase in IFN γ concentration in the supernatant, despite elevated *IFNG* mRNA levels and increased CD25 expression, raises questions about the functional impact of cholesterol treatment. We discussed this discrepancy on page 6 as follows:

“While we observed increased CD25 expression and elevated *IFNG* and *TNF* mRNA levels, the concentration of IFN γ in the supernatant did not significantly increase in cholesterol-enriched media and those of TNF α were only slightly higher. This discrepancy suggests that while cholesterol may enhance the transcriptional activation of *IFNG* and *TNF*, it does not necessarily translate to increased protein production, possibly due to post-transcriptional or translational regulatory mechanisms.”

Although the functional impact of cholesterol on IFN γ secretion may appear limited based on our current data, the observed changes in *IFNG* mRNA and CD25 expression might still be indicative of cholesterol's role in modulating T cell activation and differentiation. These findings suggest that cholesterol may influence T cell responses in ways that are not solely dependent on IFN γ secretion, potentially affecting other aspects of T cell function or interacting with other signaling pathways. Further studies are needed to explore these possibilities and to determine the broader implications of cholesterol's role in T-cell biology.

We discussed this on page 8 as follows:

“While we observed changes in *IFNG* mRNA levels, further studies are warranted to determine whether these transcriptional changes translate into some functional outcomes, such as cytokine production, T cell-mediated immune responses *in vivo*, or interaction with other signaling pathways.”

5. Not only did cholesterol in the media induce IFNG mRNA, it also induced TNF mRNA

and there was some detectable TNF in the supernatant. TNF is insufficiently considered in this manuscript.

REPLY: We agree that the increase in *TNF* mRNA and TNF concentrations in the supernatant was noteworthy. However, we chose not to focus extensively on TNF because it is not strictly associated with the Th1 response, and the mRNA increase was less significant compared to *IFNG*. Additionally, the rise in TNF concentrations was only marginal.

However, we edited the discussion as follows:

“While we observed increased CD25 expression and elevated *IFNG* and *TNF* mRNA levels, the concentration of IFN γ in the supernatant did not significantly increase in cholesterol-enriched media and those of TNF α were only slightly higher. This discrepancy suggests that while cholesterol may enhance the transcriptional activation of *IFNG* and *TNF*, it does not necessarily translate to increased protein production, possibly due to post-transcriptional or translational regulatory mechanisms. Bulk RNA sequencing revealed differential gene expression in pathways including AP-1 and SGK1. SGK1 emerged as a potential contributor, as its inhibition led to a partial decrease in *IFNG* expression with a similar trend observed for *TNF*, suggesting a role for this kinase in modulating the transcriptional response to cholesterol.”

“Interestingly, cholesterol supplementation also led to an increase in *TNF* mRNA levels, accompanied by a slight increase in TNF α concentrations in the supernatant. Additionally, the increase in *TNF* mRNA was attenuated when the cells were cultured with an SGK1 inhibitor. These findings suggest that cholesterol, through activation of SGK1, may also enhance TNF α production by promoting the Th1 function (Kleinewietfeld et al., 2013). The increase in TNF α concentrations could also be mediated through the activation of other downstream signaling pathways, such as NF- κ B, which plays a pivotal role in cytokine expression (Liu et al., 2017). The involvement of NF- κ B underscores the potential mechanistic link between cholesterol metabolism and immune modulation, indicating that cholesterol may act as an upstream regulator of pro-inflammatory responses by influencing key transcription factors.”

6. The PCA plot in Supplement Figure 1C is not convincing. In both groups, two of the three samples are very similar, but one sample differs significantly from the other two in gene expression, making the validity of the analysis questionable.

REPLY: We agree with the reviewer that there was heterogeneity within donors. This is one of the reasons why we conducted additional qPCR to confirm increased gene expressions, finally revealing similarity across donors.

7. That SGK1 mediates the effects of membrane cholesterol has long been known (PMID: 19910701). Do the authors know to which extent the cholesterol in the medium alters membrane composition and to which extent it is metabolized by the cells?

REPLY: We agree with the reviewer that the effect of membrane cholesterol on SGK1 was previously reported. However, it was not fully described that SGK1 activation could lead to Th1 differentiation in the context of supplementation of CD4⁺ memory T cells with cholesterol. We clarified our objective on page 4 as follows:

“Here, we examined human CXCR3⁺ CD4 memory T cells isolated from the peripheral blood of healthy donors to determine whether and how cholesterol plays a role in driving Th1 homeostasis, including the expression of IFN γ .”

We did not investigate directly the effect of cholesterol on the membrane composition. However, we investigated the effects of methyl- β -cyclodextrin (mbcd), a well-established cyclic oligosaccharide commonly employed to deplete cholesterol from the cellular membrane. The cells cultured in media supplemented by mbcd did not show any increase of *IFNG* mRNA suggesting the role of the membrane cholesterol in the observed effect.

This is now described on page 5 and figure 2 as follows:

Impact of methyl- β -cyclodextrin

To differentiate the effects of cholesterol from those of the methyl- β -cyclodextrin (mbcd), and to assess the role of cholesterol in plasma membranes in regulating Th1 function, we cultured CD4⁺ effector/memory cells with or without mbcd, a well-established cyclic oligosaccharide commonly used to deplete cholesterol from cellular membranes. The *IFNG* expression remained unchanged in the presence of mbcd ($p=0.81$), whereas a 45-fold increase in *IFNG* expression was observed in the presence of cholesterol (Figure 2A). Similarly, there was no difference in the concentrations of IFN γ or TNF α in the supernatant of the cells at 72 hours (Figure 2B).

8. Cholesterol treatment induces a subtle state of activation (CD25, IFNG mRNA, TNF mRNA). Is the activation threshold of a T cells dependent on the membrane cholesterol content? The authors should provide data supporting this.

REPLY: We appreciate the reviewer's insightful question regarding the relationship between membrane cholesterol content and the activation threshold of T cells. While our current study focuses on the effects of cholesterol enrichment on CD4⁺ memory T cell activation and cytokine expression, we did not directly measure changes in the activation threshold of T cells relative to their membrane cholesterol content. However, as explained above, we demonstrated that the depletion of the cholesterol membrane did not lead to any increase in *IFNG* mRNA.

Additionally, as discussed in the introduction, existing literature suggests that cholesterol-rich lipid rafts play a critical role in TCR signaling and the formation of immunological synapses, which could influence the activation threshold. To address this question more thoroughly, future experiments specifically designed to assess the activation threshold in relation to varying cholesterol levels in the membrane would be valuable. We will consider this in our ongoing and future studies. We discussed this on page 8 as follows:

“Additionally, further research exploring whether the activation threshold of T cells is dependent on the membrane cholesterol content would be valuable.”

September 24, 2024

RE: Life Science Alliance Manuscript #LSA-2024-02890-TR

Aurelie Hanin
Yale

Dear Dr. Hanin,

Thank you for submitting your revised manuscript entitled "Cholesterol Promotes IFNG mRNA Expression in CD4+ effector/memory Cells by SGK1 Activation". We would be happy to publish your paper in Life Science Alliance pending final revisions necessary to meet our formatting guidelines.

- please be sure that the authorship listing and order is correct
- please add the author contributions to the main manuscript text
- please consult our manuscript preparation guidelines <https://www.life-science-alliance.org/manuscript-prep> and make sure your manuscript sections are in the correct order
- please use the [10 author names, et al.] format in your references (i.e. limit the author names to the first 10)
- please upload your table files as editable doc or excel files

A. FINAL FILES:

B. MANUSCRIPT ORGANIZATION AND FORMATTING:

Thank you for your attention to these final processing requirements. Please revise and format the manuscript and upload materials within 5 days.

Sincerely,

September 26, 2024

RE: Life Science Alliance Manuscript #LSA-2024-02890-TRR

Aurelie Hanin
Yale University
Neurology
300 George Street
New Haven, CT 06511

Dear Dr. Hanin,

Thank you for submitting your Research Article entitled "Cholesterol Promotes IFNG mRNA Expression in CD4+ effector/memory Cells by SGK1 Activation". It is a pleasure to let you know that your manuscript is now accepted for publication in Life Science Alliance. Congratulations on this interesting work.

DISTRIBUTION OF MATERIALS:

Again, congratulations on a very nice paper. I hope you found the review process to be constructive and are pleased with how the manuscript was handled editorially. We look forward to future exciting submissions from your lab.

Sincerely,
